# Anthropogenic Disturbances Have Contributed to Degradation of River Water Quality in Arid Areas

**Li Ji [1,\*], Yuan Li [1,2], Guixiang Zhang [1] and Yonghong Bi [2,\*]**

[1] School of Environmental Science and Engineering, Taiyuan University of Science and Technology, Taiyuan 030024, China; liyuan_198711@sina.com (Y.L.); zhanggx@tyust.edu.cn (G.Z.)

[2] State Key Laboratory of Freshwater Ecology and Biotechnology, Institute of Hydrobiology, Chinese Academy of Sciences, Wuhan 430072, China

\* Correspondence: jili@tyust.edu.cn (L.J.); biyh@ihb.ac.cn (Y.B.);
Tel.: +86-351-696-2589 (L.J.); +86-27-6878-0016 (Y.B.)

**Abstract:** The earth has been reshaped for millennia. The accelerating pace of anthropogenic activities has generated enormous impacts on the water environment. As one of the main drivers of landscape change, anthropogenic disturbance has brought many negative effects on rivers. Studying the relationship between anthropogenic disturbances and river water quality is of significance for regional conservation and ecosystem management, while the relationship remains poorly understood in the current. In this study, we quantified anthropogenic disturbances by introducing the concept of the hemeroby index and evaluated rivers' water quality in eight sub-watersheds on the Loess Plateau. The results indicated that 37.5% of the sub-watersheds were in Eutrophic status, and 62.5% were in Marginal water quality index. The river water quality was most poor in the southwestern region near the Yellow River with high-level anthropogenic disturbance. A correlation analysis between water quality indicators and hemeroby suggested that anthropogenic disturbance contributed to a significant water quality deterioration trend ($p < 0.01$). The river water quality was relatively sensitive to the changes of completely disturbed land-use covers, including urban and industrial land. Our findings provide theoretical guidance for regional water resources conservation and ecosystem management in arid areas.

**Keywords:** hemeroby; land-use covers; the Loess Plateau; trophic state; water quality



## Highlights

1. A method to quantify human disturbance on the landscape was developed;
2. The rivers in Eastern Loess Plateau were relatively sensitive to construction land;
3. The rivers near the Yellow River are more vulnerable affected by soil erosion;
4. Managers should adopt measures to protect water resources by land-use planning.

## 1. Introduction

Intensive anthropogenic activities and rapid economic development are likely to lead to a vast increase in water resource use, and eventually, a water crisis in the long run [1]. On the other hand, anthropogenic activities (excessive pollution discharge, nutrient pollution from increased construction land, agricultural surface source pollution, and the industry discharge, including textile industry, metal mining industry, and pharmaceutical industry, etc.) [2–4] have led to irreversible changes in landscape structure, which have generated enormous impacts on river ecosystems [5]. The deterioration of water quality has constrained socio-economic development, even posing a severe threat to the ecosystem, food safety, and human health [6–8], which has become a common problem facing all countries in the world [9]. Therefore, an urgent need exists for information to help us better understand the ecological effects of human activities and to reduce their negative effects on the landscape.

Anthropogenic activities, including changes in the land-use type and vegetation cover, increase in fragmentation, destruction of biological communities, and pollution of air, soil, and water, have increased worldwide and profoundly altered the natural states of the aquatic ecosystem and their surrounding environment [10,11]. Furthermore, urban waste discharge and agricultural non-point source pollution substantially increased loadings of nutrients and organic pollutants into waters, thereby leading to eutrophication and deteriorating water quality [12,13]. One study found that anthropogenic fertilizer inputs and wastewater discharge were the leading cause of eutrophication upstream of the Three Gorges Reservoir [14], and the surface water quality was closely related to land-use type [15]. Another study analyzed the relationship between anthropogenic activities and hydrochemical indices of the Fen River Basin; its results demonstrated that primary pollution sources were related to the land-use pattern of the high proportions of the cropland and the low proportions of the forest [16]. Even many microscopic pollutants, such as pharmaceuticals, organic polymers, and suspended solids, that are difficult to identify are already discharged into rivers and reservoirs [17]. In addition, land-use changes and urbanization caused by anthropogenic activity directly degraded water quality and the aquatic ecosystem [18]. Anthropogenic activity has been looked at as a critical factor for the water environment.

Currently, the intensity and scale of human transformation of nature have been increasing. More than half of the global land is disturbed by anthropogenic activities [19], which led the anthropogenic disturbance and pressure on the aquatic ecological environment to increase. Therefore, assessing the effects of anthropogenic activities with the quantitative index on water quality is of great significance for correctly understanding the scale, intensity, and spatio-temporal changes of anthropogenic activities [20]. Land-use/cover change (such as farmland expansion, afforestation, deforestation, urbanization, and industrialization) increases the vulnerability of the aquatic ecosystem [21]. Quantitative evaluation of regional human activity intensity is necessary to recognize and quantify human interference effects directly. Halpern et al. [22] calculated the impacts of human stressors on marine ecosystems globally and found that human-made pressure had seriously affected one-fifth of the world's oceans. The net primary production of the biosphere caused by human intervention has risen from 6.9 in 1910 to 14.8 GtC/year in 2005 [23]. Quantitative evaluation of regional anthropogenic activities would help us obtain insight into impacts from humans. A correlation analysis between landscape indices and hemeroby suggested that the landscape patterns in regions with high-level human disturbance were relatively sensitive to water quality and species richness [24]. The findings in this paper provide additional spatial information and theoretical guidance.

The degree of anthropogenic disturbance expressed as the hemeroby index (HI), was used to evaluate the impact of anthropogenic activities based on remote sensing, socioeconomic or ground survey data to quantify the level of anthropogenic disturbance in a specific area [25,26]. HI converts land-use types into values representing the degrees of anthropogenic disturbance, and a higher value indicates a strong disturbance on the land-use type or ecosystem [27]. Yang and Song [28] assessed the spatio-temporal characteristics of ecological vulnerability (sensitivity and recovery ability of ecosystems to external disturbances) based on the HI and found that the ecological environment in high-disturbance areas was more vulnerable than low disturbance areas. Wale and Stein [29] incorporated hemeroby into a land-use monitoring system. The results on hemeroby of several time-cuts can estimate the cumulative impact of land-use changes on the environmental status. Zhao et al. [30] indicated that land use and hydrological variables could explain more than 50% of the variation in water quality, of which urban and industrial accounted for more than 70%. Although HI has been used as the quantitative intensity of anthropogenic activity index in a different watershed and proved a valuable parameter, few studies focused on the spatial relationship between HI and ecological effects of anthropogenic disturbance, especially in the arid areas.

Water ecosystem-based management needs a better understanding of water quality variations and their driving factors. Quantitative effects of anthropogenic activity on water quality are essential and urgent necessities. However, current research rarely addresses the quantified impacts of human activities on the aquatic environment at a large scale. To address this issue, we developed a strategy to quantify the impact of human activities on surface water through a case study of a typical arid area, the Loess Plateau of northern China. We analyzed the characteristics of different land-use covers and quantified the intensity of anthropogenic disturbance using remote sensing data and geographical information system technology. Then, we evaluated the eutrophication status and comprehensive index of surface water in different watersheds under different anthropogenic disturbance backgrounds.

## 2. Materials and Methods

### 2.1. Study Area

Shanxi province is located on the Loess Plateau in northern China. It covers an area of 156,700 km$^2$ between 110.23° E–114.54° E, 34.56° N–40.7333° N (Figure 1a). Shanxi province, the largest coal distribution region in China, is a typical arid area with an annual average precipitation of 350–600 mm. The agricultural and construction areas account for 36.87% (57,782 km$^2$) and 5.56% (8317 km$^2$), respectively (Table S1). Data on land-use change were collected from Resource Environment Data Cloud Platform (http://www.resdc.cn/ (accessed on 2 October 2021)) and was accurate to 1 km. We classified the land-use covers into 9 types: paddy land, dry land, green land (forest and grassland), human-made penstock, lake, marsh, construction land, saline, and alkaline land, and bare land base on the land-use-type parameters of the hemeroby model (Figure 1b). The classification of land-use cover and hierachy with respect to the HI is shown in Table S1.

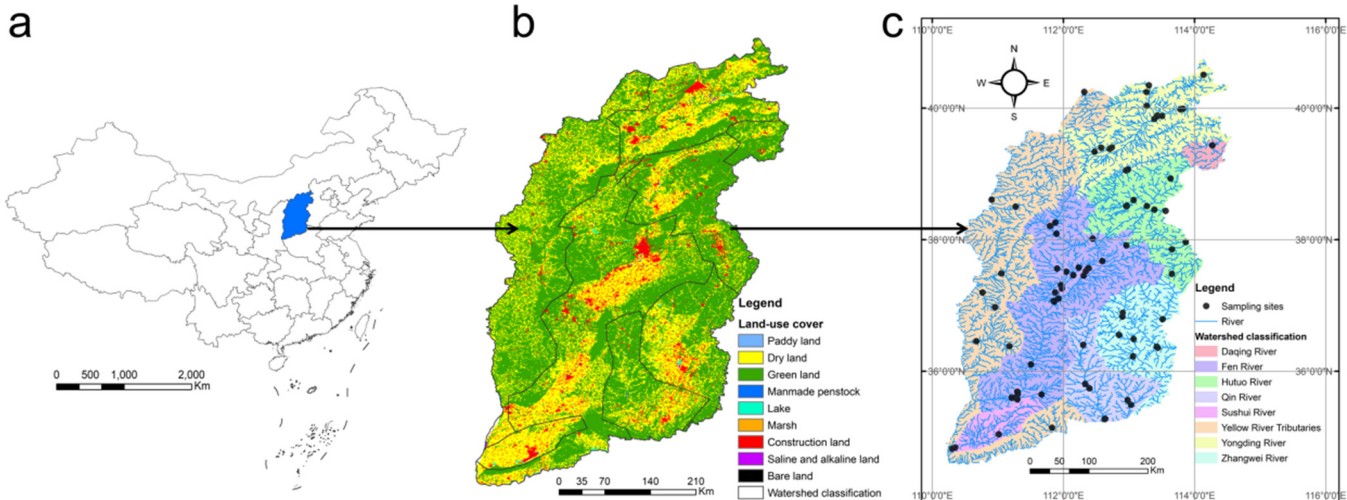

**Figure 1.** Location (**a**), land-use cover pattern (**b**), and watershed delineation (**c**) in Shanxi Province.

Shanxi Province has more than 1000 large and small rivers with noticeable seasonal variation in water quantity and 250 rivers with a drainage area of more than 100 km$^2$ (http://www.shanxi.gov.cn/ (accessed on 2 October 2021)). Shanxi's rivers originate from the mountains of the eastern and western plateaus. The rivers flowing westward to the South belongs to the Yellow River system (97,138 km$^2$, 62%), including Fenhe River, Qinhe River, and Sushui River, and the rivers flowing eastward belongs to the Haihe River system, including Hutuo River, Yongding River, Zhangwei River, and Daqing River (59,133 km$^2$, 38%) (Figure 1c). The heterogeneity and characteristics of the rivers on the arid Loess Plateau make the area an ideal region to verify the method proposed in this paper.

### 2.2. Sampling Strategy

2.2.1. Sample Collection

A total of 84 surface water samples were collected from two watersheds and eight sub-watersheds in Eastern Loess Plateau during August of 2020 (Figure 1). Water samples were collected and put into clean 5.0 L polyethylene buckets and filtered using precleaned 0.45 μm GF/C membranes. All samples were stored at 4 °C before analysis.

2.2.2. Sample Determination

Hydrochemical parameters, including total phosphorus (TP), total nitrogen (TN), chemical oxygen demand (COD), transparency (Tr), chlorophyll a (chla), manganese (Mn), copper (Cu), zinc (Zn), fluoride (F$^-$), chlorine (Cl$^-$), arsenic (As), sulfate (SO$_4^{2-}$), nitrate (NO$^{3-}$), ammonia nitrogen (NH$_4$-N) and dissolved oxygen (DO) (Table S2) were determined according to the standard methods for examining water and wastewater [31].

### 2.3. Assessment Model for River Quality

The model construction process for analyzing rivers' states based on land-use gradient and anthropogenic interference is shown in Figure 2. The specific methods and equations in the flowchart were listed as follows.

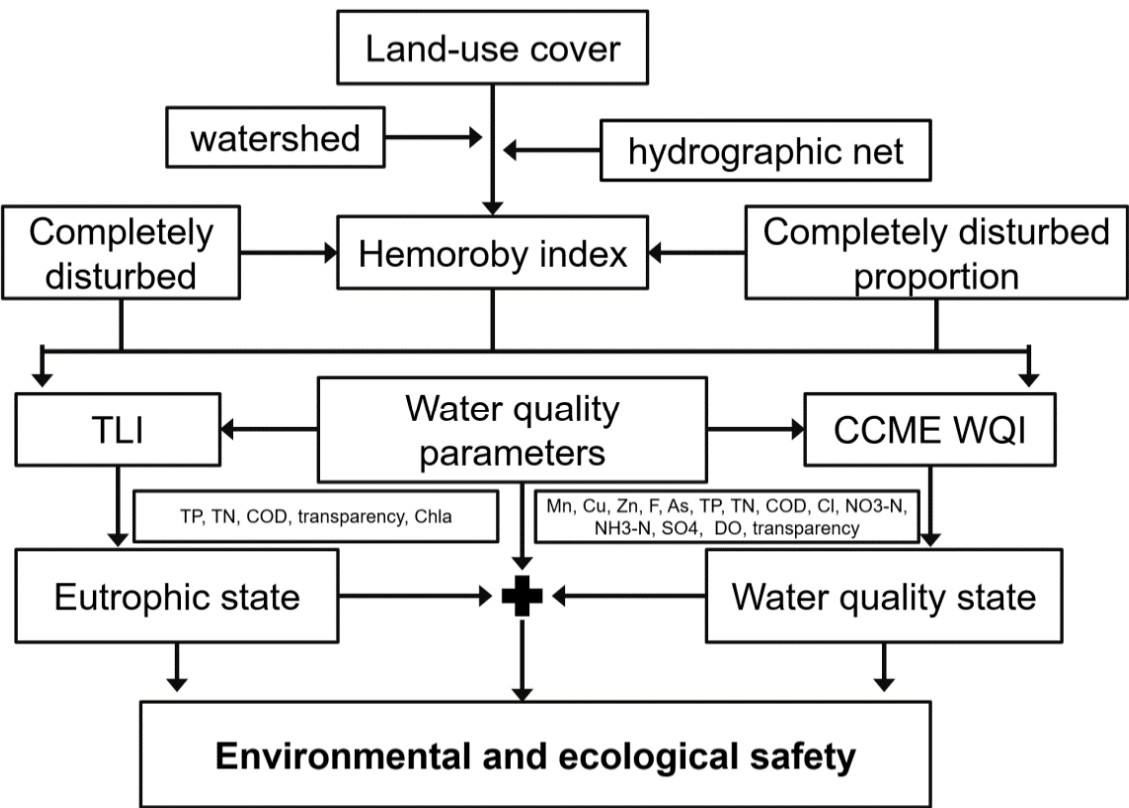

**Figure 2.** Flowchart of the model construction process for analyzing rivers' states based on land-use gradient and anthropogenic interference.

2.3.1. Eutrophication State of Rivers

The Carlson's Trophic Level Index (*TLI*) is the most popular and important method to reflect the eutrophication degree of the waterbody [32]. The TLI of each index was calculated using Equations (1)–(5). The TLI was determined using Carlson's [33] method, as follows Equations (6) and (7).

$$TLI(chla) = 10 \times (2.5 + 1.086 \times \ln chla) \tag{1}$$

$$TLI(TP) = 10 \times (9.436 + 1.624 \times \ln TP) \tag{2}$$

$$TLI(TN) = 10 \times (5.453 + 1.694 \times \ln TN) \tag{3}$$

$$TLI(SD) = 10 \times (5.118 - 1.94 \times \ln SD) \tag{4}$$

$$TLI(COD) = 10 \times (0.109 + 2.661 \times \ln COD) \tag{5}$$

$$TLI = \sum_{j=1}^{m} W_j \times TLI_j \tag{6}$$

$$W_j = \frac{r_{ij}^2}{\sum_{j=1}^{m} r_{ij}^2} \tag{7}$$

where $TLI_j$ is the Trophic Level Index of the $j$th parameter; $W_j$ is the correlation weight of the trophic level index of the $j$th parameter; $r_{ij}$ is the correlation coefficient between the $j$th parameter and the reference parameter of chla; $m$ is the number of evaluation parameters.

The calculation results of $W_j$ were performed according to Li et al.'s method [34]. The *TLI* of rivers was classified by a series of continuous numbers from 0 to 100: $TLI \leq 30$ Oligotropher; $30 \leq TLI \leq 50$ Mesotropher; $TLI > 50$ Eutropher; $50 < TLI \leq 60$ Light Eutropher; $60 < TLI \leq 70$ Middle Eutropher; $TLI > 70$ Hyper Eutropher.

### 2.3.2. Water Quality Assessment

Water quality index (WQI) is a kind of environmental quality index. It is a combination of several water quality parameters of the test results to describe the quality of water quality of a comprehensive dimensionless value. The Canadian Council of Ministers of the environment water quality index (*CCME WQI*) is a tool for measuring the pollution status of a waterbody from three aspects: quantity, frequency, and extent of pollutants exceeded water quality standards which reflect the comprehensive situation of the water environment succinctly and intuitively [35]. In our study, we selected 14 parameters except for ChlorophyII a to calculate the *CCME WQI* index. The desired criteria values of each parameter were based on Class III guidelines of China environmental quality standards for surface water ("Class III" for short, GB3838-2002). According to water quality and measured value guidelines, the *CCME WQI* score ranges from 0 to 100.

*CCME WQI* was calculated through Equation (8).

$$CWQI = 100 - \left[ \frac{\sqrt{(F1^2 + F2^2 + F3^2)}}{1.732} \right] \tag{8}$$

where the scope (*F1*) is the number of parameters that were not compliant with the water quality guidelines; the frequency (*F2*) is the number of times that the guidelines were not respected; the amplitude (*F3*) is the difference between non-compliant measurements and the corresponding guidelines [36].

*F1*, *F2*, and *F3* were calculated through Equations (9)–(13). For parameters in which higher values denoted poor quality (such as total nitrogen), the excursion (e) was calculated as Equation (11). For parameters in which higher values denoted better quality (such as the presence of DO), the excursion (e) was calculated as Equation (12).

$$F1 = \left( \frac{\text{Number of failed parameters}}{\text{Total number of parameters}} \right) \times 100 \tag{9}$$

$$F2 = \left( \frac{\text{Number of failed results}}{\text{Total number of results}} \right) \times 100 \tag{10}$$

$$e_{ij} = \left( \frac{\text{Failed test result}_{ij}}{\text{Guideline}_{ij}} \right) \times 100 \tag{11}$$

$$e_{ij} = \left( \frac{\text{Guideline}_{ij}}{\text{Failed test result}_{ij}} \right) \times 100 \tag{12}$$

$$F3 = \frac{100 \sum\limits_{i=1}^{M} \sum\limits_{j=1}^{N} e_{ij}}{\sum\limits_{i=1}^{M} \sum\limits_{j=1}^{N} e_{ij} + MN} \tag{13}$$

where $N$ is the total number of pollutants; $M$ was the total number of the measured $i$th pollutant. $i$ and $j$ indicate the concentration of the $i$th pollutant of the $j$th time measured in assimilated water. The *CCME WQI* of rivers was classified by a series of continuous numbers from 0 to 100: *CCME WQI* < 45 Poor; 45 ≤ *CCME WQI* < 65 Marginal; 65 ≤ *CCME WQI* < 80 Fair; 80 ≤ *CCME WQI* < 95 Good; *CCME WQI* ≥ 95 Excellent [37].

### 2.4. Evaluation Model for Anthropogenic Interference

Hemeroby index (HI) was calculated by Equation (14) expressing the human disturbance index; $n$ is the number of land-use covered in the statistical unit; $S_i$ is the area of the current land-use type; $S$ is the total area of the statistical units. The HI for each land-use type is listed in Table S1.

$$HI = \sum_{i}^{n} \frac{S_i}{S} \times H_i \tag{14}$$

### 2.5. Data Statistics and Analyses

Remote sensing data, statistical data, and water quality data were collected and processed according to the flowchart in Figure 2. Sources of nitrogen and phosphorus nutrients in Shanxi Province were collected from China Statistical Yearbook [38].

One-Way ANOVA tested the differences (between different watersheds and sub-watersheds at 95% confidence interval ($p$-values < 0.05)) followed by Kruskal–Wallis tests based on Sigma Plot 14.0 (Systat Inc., San Jose, CA, USA) (Mahapoonyanont et al., 2010). We employed linear regression analysis to explain the correlation between the human disturbance index and the river's status calculated by Sigma Plot. The spatial distributions of eutrophication and water quality state were predicted using Inverse Distance Weighting interpolation (IDW) based on ArcGIS 10.2 (ESRI Inc., West Redlands, CA, USA) [39].

## 3. Results

### 3.1. Spatial Variation of Hydrochemical Indices

The spatial variations of 15 hydrochemical indices are shown in Table 1. The water of the sub-watershed was very turbid. It was found that the mean concentrations of TN (>1.0 mg L$^{-1}$) exceeded the Class III standard for surface water (shorter form Class III). The TN concentration in decreasing order was: Yellow River (7.75 mg L$^{-1}$) > Shanxi Province (6.98 mg L$^{-1}$) > Hai River (5.98 mg L$^{-1}$). The maximum of TN was found in Yellow River Tributaries, reaching 8.73 mg L$^{-1}$. The TP exceeded Class III (>0.1 mg L$^{-1}$) in the Yellow River Tributaries (0.13 mg L$^{-1}$) and the Yongding River (0.202 mg L$^{-1}$). The TP concentration in decreasing order was Hai River (0.13 mg L$^{-1}$) > Shanxi Province (0.10 mg L$^{-1}$) > Yellow River (0.08 mg L$^{-1}$). For other hydrochemical indices, the concentrations of COD, SO4$^{2-}$, and NH$_4$-N exceeded Class III sporadically. There was 37.5% of the sub-watersheds with Eutrophic status, and 62.5% were in the Marginal water quality index.

**Table 1.** Sampling numbers and mean concentration of 15 hydrochemical indices in different watersheds and rivers.

| Watershed | River | Sampling Numbers | Mean Concentrations (mg L$^{-1}$) | | | | | | | | | | | | | | |
|---|---|---|---|---|---|---|---|---|---|---|---|---|---|---|---|---|---|
| | | | TP | TN | COD | Tr (cm) | Chla | Mn | Cu | Zn | F$^-$ | Cl$^-$ | As | SO4$^{2-}$ | NO3$^-$ | NH4-N | DO |
| Yellow River | Fen River | 28 | 0.08 | 8.37 | 4.16 | 33.77 | 0.0350 | 0.0131 | 0.0046 | 0.0070 | 0.71 | 157.31 | 0.0012 | 286.82 | 6.88 | 0.69 | 8.35 |
| | Yellow River Tributaries | 9 | 0.13 | 8.73 | 2.60 | 32.20 | 0.0340 | 0.0007 | 0.0042 | 0.0010 | 0.61 | 64.60 | 0.0023 | 109.20 | 6.20 | 1.44 | 9.69 |
| | Qin River | 7 | 0.02 | 3.84 | 1.72 | 37.69 | 0.0290 | 0.0032 | 0.0025 | 0.0042 | 0.48 | 76.17 | 0.0016 | 209.87 | 2.28 | 0.17 | 11.28 |
| | Sushui River | 3 | 0.10 | 8.19 | 8.87 | 44.28 | 0.0352 | 0.0011 | 0.0179 | 0.0015 | 0.66 | 147.70 | 0.0025 | 262.00 | 0.51 | 0.53 | 9.66 |
| | Total | 47 | 0.08 | 7.75 | 3.80 | 34.72 | 0.0332 | 0.0101 | 0.0054 | 0.0058 | 0.67 | 140.07 | 0.0015 | 262.37 | 5.81 | 0.75 | 9.13 |
| Hai River | Hutuo River | 10 | 0.06 | 5.57 | 2.34 | 27.22 | 0.0386 | 0.0043 | 0.0030 | 0.0021 | 0.51 | 44.98 | 0.0013 | 182.00 | 3.93 | 0.31 | 9.59 |
| | Yongding River | 16 | 0.22 | 7.35 | 3.90 | 26.92 | 0.0425 | 0.0190 | 0.0042 | 0.0035 | 0.82 | 125.05 | 0.0024 | 194.57 | 3.16 | 1.15 | 9.77 |
| | Zhangwei River | 9 | 0.05 | 3.84 | 3.43 | 30.62 | 0.0414 | 0.0077 | 0.0031 | 0.0173 | 0.50 | 94.68 | 0.0013 | 151.52 | 3.16 | 0.23 | 9.74 |
| | Daqing River | 2 | 0.01 | 7.16 | 0.75 | 30.92 | 0.0306 | 0.0412 | 0.0034 | 0.0082 | 0.52 | 98.36 | 0.0014 | 135.80 | 3.02 | 0.03 | 10.77 |
| | Total | 37 | 0.13 | 5.98 | 3.26 | 28.06 | 0.0385 | 0.0129 | 0.0036 | 0.0068 | 0.67 | 100.20 | 0.0019 | 180.59 | 3.32 | 0.66 | 9.76 |
| Shanxi Province | | 84 | 0.10 | 6.98 | 3.53 | 31.81 | 0.0323 | 0.0113 | 0.0046 | 0.0062 | 0.67 | 123.24 | 0.0016 | 227.84 | 4.76 | 0.70 | 9.41 |

### 3.2. Hemeroby Index of Different Sub-Watersheds

Hemoroby index, completely disturbed, and completely disturbed proportion of the sub-watersheds can be seen in Table 2. The results showed that the hemoroby index (0.7036), completely disturbed (0.1074), and completely disturbed proportion (15.27%) of Sushui River were maximum. The HI (0.5832), completely disturbed (0.0082), and completely disturbed proportion (1.40%) of Daqing River (0.6418) were minimum. The spatial distribution of these three indicators was high in the southwest and low in the northeast of Shanxi Province (Figure 3).

**Table 2.** Anthropogenic interference levels, including Hemoroby index, completely disturbed, and completely disturbed proportion in different watersheds and rivers.

| Watershed | River | Watershed Area (km²) | Hemoroby Index | Completely Disturbed | Completely Disturbed Proportion (%) |
|---|---|---|---|---|---|
| Yellow River | Fen River | 38,014 | 0.6218 | 0.0569 | 9.15 |
| | Yellow River tributaries | 43,105 | 0.6096 | 0.0231 | 3.79 |
| | Qin River | 12,621 | 0.6209 | 0.0479 | 7.72 |
| | Sushui River | 5540 | 0.7036 | 0.1074 | 15.27 |
| Hai River | Zhangwei River | 15,713 | 0.6323 | 0.0600 | 9.49 |
| | Daqing River | 2173 | 0.5832 | 0.0082 | 1.40 |
| | Yongding River | 22,279 | 0.6418 | 0.0701 | 10.93 |
| | Hutuo River | 17,255 | 0.6074 | 0.0515 | 8.49 |

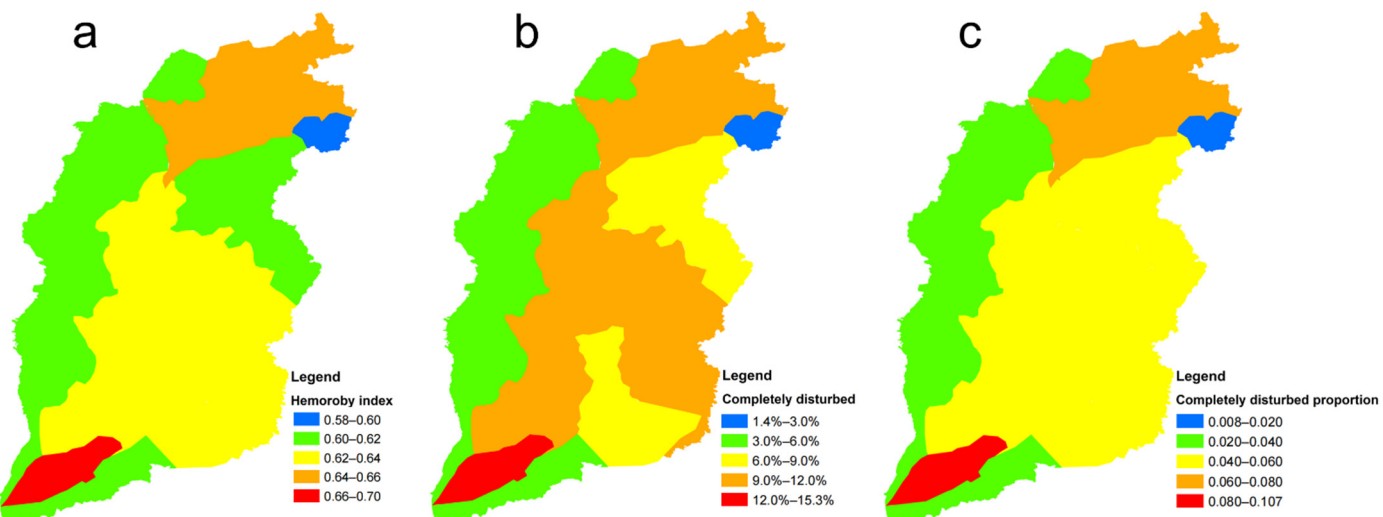

**Figure 3.** Spatial distribution of the anthropogenic interference in different rivers. (**a**) Hemoroby index; (**b**) completely disturbed; (**c**) completely disturbed proportion.

### 3.3. Rivers' State of Different Watersheds and Sub-Watersheds

3.3.1. Trophic Level Index of Different Watersheds and Rivers

Trophic level index (*TLI*) in different watersheds and sub-watersheds are shown in Table 3. The mean *TLI* of the two watersheds and Shanxi Province were Mesotropher ($30 \leq TLI \leq 50$). The order of *TLI* was: Yellow River (49.00) > Shanxi Province (48.67) > Hai River (48.24) (Figure 4). The *TLI* of Sushui River (55.00), Yongding River (51.62), and Fen River (50.39) were Eutropher (*TLI* > 50). The maximum *TLI* 68.82 was found in Yongding River in Middle Eutropher ($60 < TLI \leq 70$). We predicted the spatial pattern of the *TLI* of all rivers in Shanxi Province based on the spatial analysis model of ArcGIS (Figure 5a). The results showed that the *TLI* of all rivers were ranged from 43.07 to 54.76. The rivers' *TLI* in Eutropher state was found in the southwest and north of Shanxi Province.

**Table 3.** Results of the eutrophication state (*TLI*) and water quality (*CCME WQI*) in different watersheds and rivers.

| Watershed | River | TLI | | | | CCME WQI | | | |
|---|---|---|---|---|---|---|---|---|---|
| | | Mean | SD | Maximum | Minimum | Mean | SD | Maximum | Minimum |
| Yellow River | Fen River | 50.39 | 7.13 | 60.18 | 35.81 | 59.43 | 17.90 | 100.00 | 27.35 |
| | Yellow River Tributaries | 48.37 | 7.75 | 61.91 | 39.30 | 54.33 | 14.70 | 77.54 | 31.93 |
| | Qin River | 41.67 | 1.34 | 43.10 | 39.60 | 72.15 | 9.71 | 86.75 | 59.68 |
| | Sushui River | 55.00 | 4.83 | 60.43 | 48.71 | 54.68 | 17.65 | 74.57 | 40.89 |
| | Total | 49.00 | 7.42 | 61.91 | 35.81 | 60.04 | 16.80 | 100.00 | 27.35 |
| Hai River | Hutuo River | 44.97 | 9.39 | 57.22 | 28.38 | 67.24 | 17.91 | 100.00 | 36.27 |
| | Yongding River | 51.62 | 7.55 | 68.82 | 40.36 | 60.90 | 19.62 | 88.01 | 31.45 |
| | Zhangwei River | 47.16 | 3.29 | 53.46 | 41.46 | 76.45 | 18.60 | 100.00 | 41.87 |
| | Daqing River | 36.42 | 3.58 | 60.12 | 57.06 | 57.53 | 0.66 | 45.35 | 36.42 |
| | Total | 48.24 | 8.05 | 68.82 | 28.38 | 66.44 | 19.21 | 100.00 | 31.45 |
| Shanxi Province | | 48.67 | 7.75 | 68.82 | 28.38 | 62.76 | 17.95 | 100.00 | 27.35 |

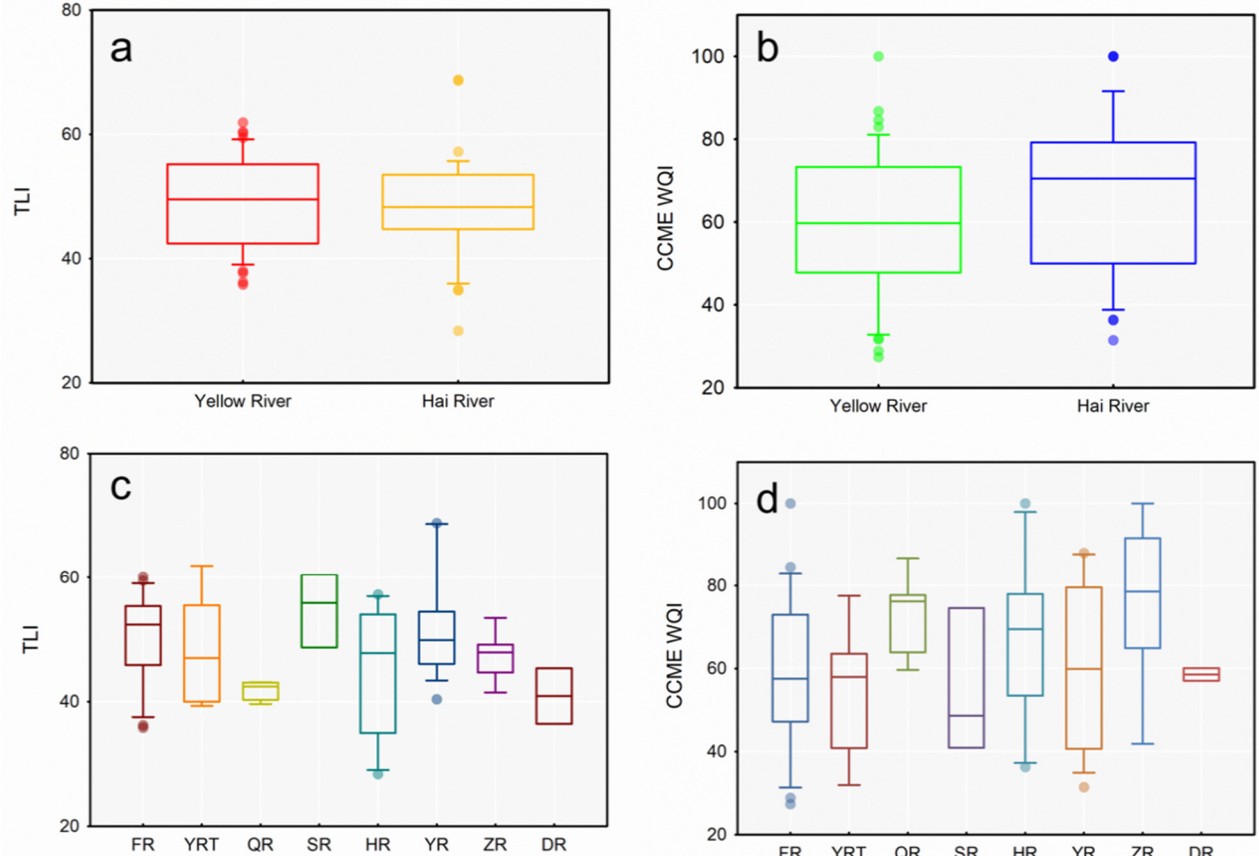

**Figure 4.** Box plots of the trophic state and water quality in different watersheds and rivers. (**a**,**c**) comparison of *TLI* index in different watersheds and rivers; (**b**,**d**) comparison of *CCME WQI* index in different watersheds and rivers. FR: Fen River, YRT: Yellow River Tributaries, QR: Qin River, SR: Sushui River, HR: Hutuo River, YR: Yongding River, ZR: Zhangwei River, DR: Daqing River.

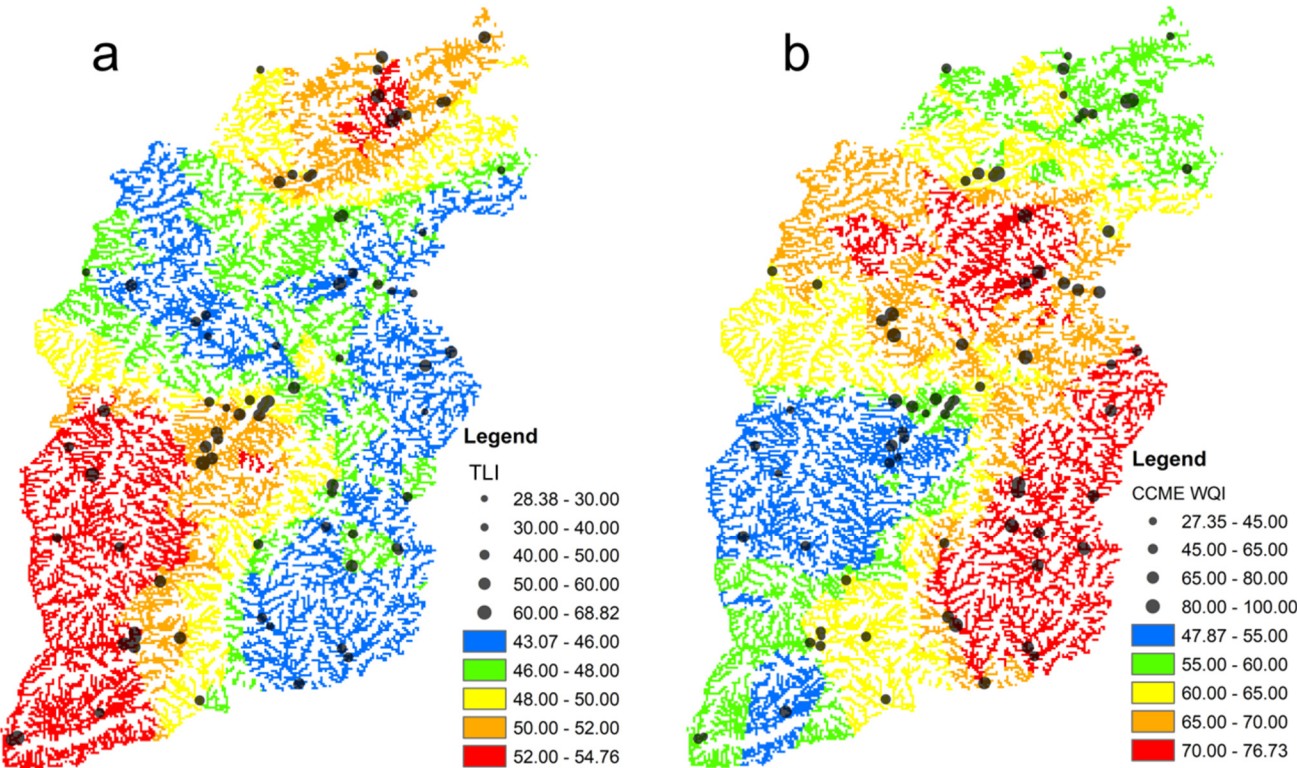

**Figure 5.** Spatial distribution of the eutrophication state and water quality in different Shanxi's rivers. (**a**) *TLI* index; (**b**) *CCME WQI* index.

### 3.3.2. *CCME WQI* of Different Watersheds and Sub-Watersheds

The water quality indexes (*CCME WQI*) of different watersheds and sub-watersheds are shown in Table 3. The mean *CCME WQI* of two watersheds and Shanxi Province were in Marginal (45 ≤ *CCME WQI* < 65) and Fair (65 ≤ *CCME WQI* < 80) state, respectively. The order of *CCME WQI* was: Hai River (66.44) > Shanxi Province (62.76) > Yellow River (60.04) (Figure 5). *TLI* of Sushui River (55.00), Yongding River (51.62), and Fen River (50.39) were Eutropher (*TLI* > 50). The mean *CCME WQI* was minimum found in Yellow River Tributaries (54.33) in the Marginal state. No Poor state (*CCME WQI* < 45) rivers were found in all sub-watersheds. We predicted the spatial pattern of the *CCME WQI* of all rivers in Shanxi Province based on the spatial analysis model of ArcGIS (Figure 5b). The results showed that the *CCME WQI* of all rivers were ranged from 47.87 to 76.73. The rivers' *CCME WQI* was found in Marginal state in the southwest and north of Shanxi Province. The rivers were in Fair state in the southeast and central part.

### 3.4. Relationship between Hemeroby Index and Rivers' Environment

#### 3.4.1. Relationship between HI and *TLI*

We analyzed the correlations between the anthropogenic interference, including the HI (Figure 6a), completely disturbed (Figure 6b), completely disturbed proportion (Figure 6c), and rivers' *TLI* in all sub-watersheds. *TLI* was positively correlated with the HI ($p < 0.01$, $r^2 = 0.88$), completely disturbed ($p < 0.0001$, $r^2 = 0.94$) and completely disturbed proportion ($p < 0.0001$, $r^2 = 0.93$). Among them, the coefficient between completely disturbed and *TLI* was the highest.

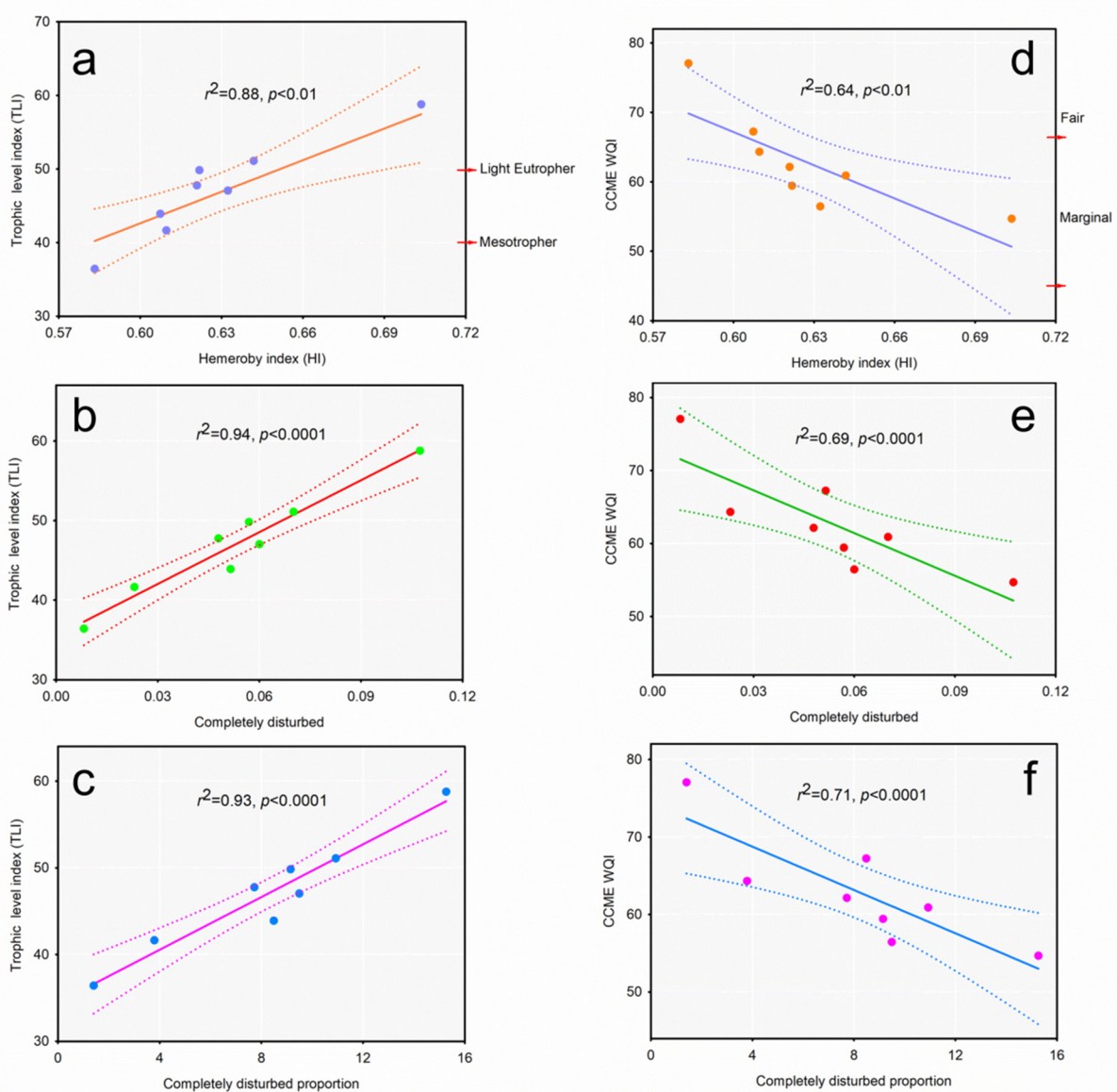

**Figure 6.** Correlations between anthropogenic interference and rivers' environment quality in Shanxi Province. The confidence interval of regression model is 95%. *p* < 0.01 indicates the regression coefficient has a significant level. (**a**) Correlations between *TLI* and Hemoroby index; (**b**) correlations between *TLI* and Completely disturbed; (**c**) correlations between *TLI* and Completely disturbed proportion; (**d**) correlations between *CCME WQI* and Hemoroby index; (**e**) correlations between *CCME WQI* and Completely disturbed; (**f**) correlations between *CCME WQI* and completely disturbed proportion; $30 \leq TLI \leq 50$ indicates mesotropher; $50 < TLI \leq 60$ indicates light Eutropher.

### 3.4.2. Correlation between Hemoroby Index and CCME WQI

We analyzed the correlations between the anthropogenic interference, including the HI (Figure 6d), completely disturbed (Figure 6e), and completely disturbed proportion (Figure 6f), and rivers' *CCME WQI* in all sub-watersheds. *CCME WQI* was positively correlated with the HI ($p < 0.01$, $r^2 = 0.64$), completely disturbed ($p < 0.0001$, $r^2 = 0.69$), and completely disturbed proportion ($p < 0.0001$, $r^2 = 0.71$). Among, the coefficient between completely disturbed proportion and *CCME WQI* was highest.

## 4. Discussions

### 4.1. Spatial Variation of Hydrochemical Indices

To meet the rapid economic development and population growth requirements, the nitrogen load has been mainly impacted by urbanization and agriculture activities [40].

Wastewater discharge, exhaust gas, and fertilizer use were the main reasons for the high nitrogen content in rivers in the Loess Plateau [41]. Nitrogen and other compound fertilizer applied $22.6 \times 10^4$ and $67.1 \times 10^4$ tons in 2020 (Figure 7) [42]. In 2017, the amounts of nitrogen from wastewater discharge and exhaust gas ($NO_x$) were $4.66 \times 10^4$ and $52.10 \times 10^4$ tons, respectively [38]. It should be noted that the total atmospheric N deposition increased from 28.34 kg N ha$^{-1}$ a$^{-1}$ in 2001–2005 to 32.31 kg N ha$^{-1}$ a$^{-1}$ in 2010–2015 in Shanxi Province [43]. Therefore, atmospheric nitrogen could also be a source of nitrogen in rivers in the Loess Plateau.

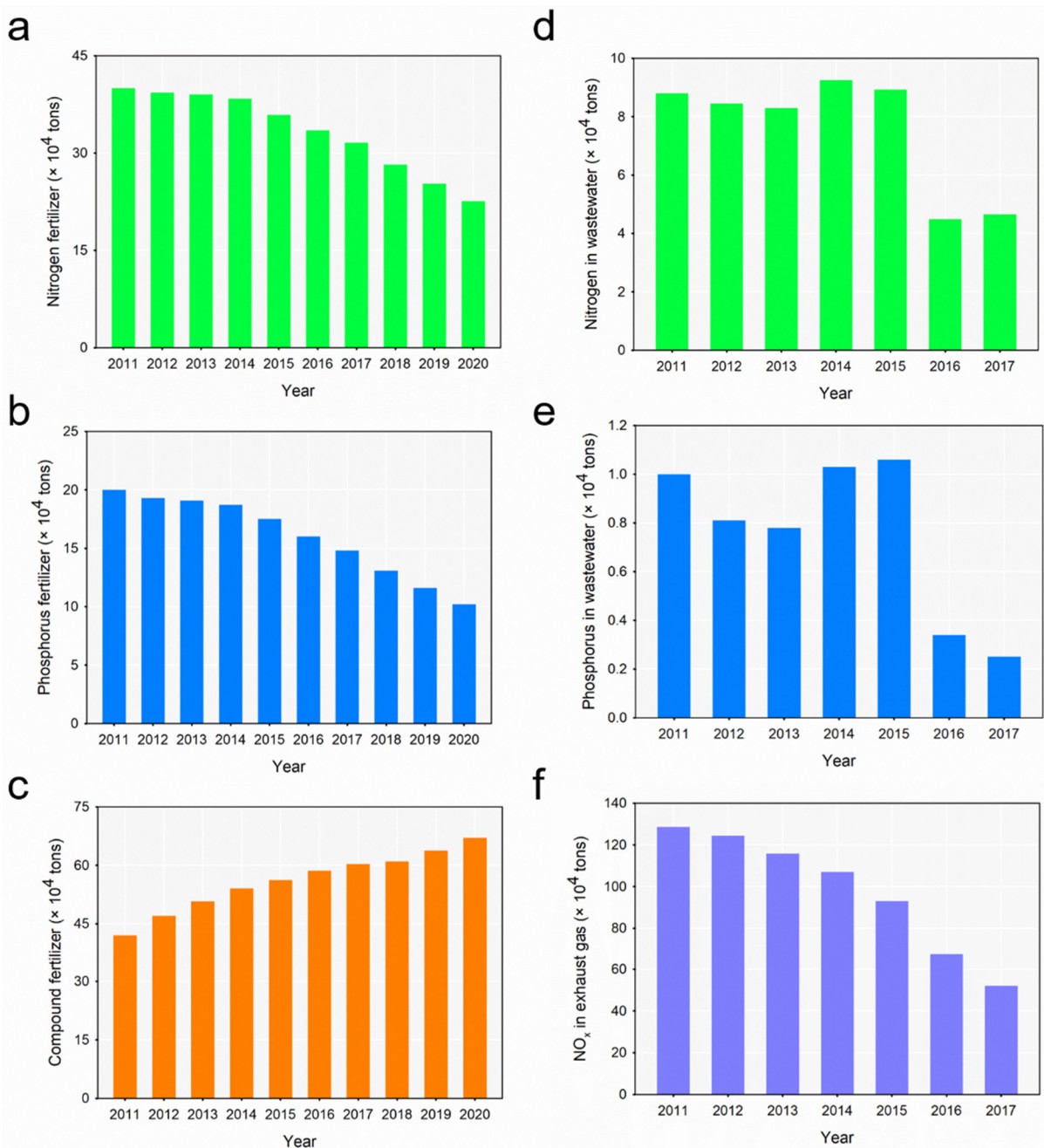

**Figure 7.** Sources of nitrogen and phosphorus nutrients in Shanxi Province. Application of nitrogen (**a**), phosphorus (**b**), and compound (**c**) fertilizer in 2011–2020; nitrogen (**d**) and phosphorus (**e**) discharge in wastewater in 2011–2017; $NO_x$ discharge in exhaust gas (**f**) in 2011–2020.

Phosphorus fertilizer application, pesticide use, and wastewater discharge are important sources of high phosphorus content in the surface water in the Loess Plateau (Li et al., 2017).

The amount of phosphorus fertilizer applied in Shanxi in 2020 reached $10.2 \times 10^4$ tons (Figure 3) [42]. The phosphorus from wastewater discharge was $0.25 \times 10^4$ tons in 2017 in the same area [38]. A large amount of unused phosphorus was transported to water bodies by farm drainage, rainwater, domestic sewage, industrial wastewater, and municipal pipeline water. Moreover, the heavy water and soil erosion in the Loess Plateau intensified the input and transportation of phosphorus nutrients [44]. Excess nitrogen and phosphorus are enriched in surface water, which could negatively affect water quality and cause eutrophication.

### 4.2. Anthropogenic Disturbances of Different Sub-Watersheds

The intensity of human interference was generally stronger in the southwest and northeast of the Shanxi Province. This heterogeneity supplied clearer details of anthropogenic activities. The east and west sides of Shanxi are mountainous areas, and the center area comprises six large basins and the Fen River Valley. Desirable locations and fertile lands mean that the low altitude area gradually became a populated center with increased economic activities [45]. The intensive anthropogenic activities have been identified as a key driver for anthropogenic land-use cover change, mainly agricultural and construction land [46].

Different landscape configurations would produce different ecological effects depending on how they were allotted [47]. The disturbance caused by anthropogenic activity would have different impacts based on landscape configuration differences. In the calculation process of HI, each land-use category was assigned a specific value that represents the exposure to environmental threats [48]. The anthropogenic interference of construction (0.99), bare (0.72), and agricultural land (0.70) were higher than other land-use covers. Because the proportions of agricultural (72%) and construction land (11%) in the Sushui River were higher than other sub-watersheds, the HI of Sushui River was the highest (0.7036).

### 4.3. Rivers' State of Different Watersheds and Sub-Watersheds

Fen River, located in the east of the Loess Plateau, was in Eutropher state. The high input of nutrients to rivers was the main reason for eutrophication. As the economic center of Shanxi Province, approximately 45% of the population, gross domestic product (GDP), and grain output are from these areas [49]. Moreover, the Fen River plays an important role in Shanxi's industrial and domestic water supply. High nitrogen content was discharged from wastewater, exhaust gas, and fertilizer use [41]. Soil erosion is an important driver of nutrient loss, including nitrogen and phosphorus, decreasing water availability [50]. Han et al. [51] showed that soil erosion in Fen River Basin exceeded $8000 \, \text{t km}^{-2} \, \text{a}^{-1}$ accounted for 15.8% of the Shanxi Province. Anthropogenic activities led the eutrophication in the Fenhe River.

The water quality indexes (*CCME WQI*) of different watersheds and sub-watersheds are shown in Table 3. The mean *CCME WQI* of two watersheds and Shanxi Province were in Marginal ($45 \leq CCME \, WQI < 65$) and Fair ($65 \leq CCME \, WQI < 80$) state, respectively. The order of *CCME WQI* was: Hai River (66.44) > Shanxi Province (62.76) > Yellow River (60.04) (Figure 5). *TLI* of Sushui River (55.00), Yongding River (51.62), and Fen River (50.39) were Eutropher (*TLI* > 50). The mean *CCME WQI* was minimum found in Yellow River Tributaries (54.33) in the Marginal state. No Poor state (*CCME WQI* < 45) rivers were found in all sub-watersheds. We predicted the spatial pattern of the *CCME WQI* of all rivers in Shanxi Province based on the spatial analysis model of ArcGIS (Figure 6b). The results showed that the *CCME WQI* of all rivers were ranged from 47.87 to 76.73. The rivers' *CCME WQI* was found in Marginal state in the southwest and north of Shanxi Province. The rivers were in Fair state in the southeast and central part.

Fen River Basin receives industrial wastewater and domestic sewage from the surrounding areas. Only 57.8% of the hydrochemical indices in the main tributaries reached a Class III surface water standard, such as Lan River and Jian River [52]. Taiyuan City is the capital of Shanxi Province and the center of Fenhe River Basin. In 2017, industrial wastewater and urban domestic sewage discharge were 37.39 and 265.2 million tons, re-

spectively [38]. The surrounding coking plants, fertilizer plants, and roads that mainly served for coal transport brought many pollutants to the rivers [53]. Eutrophication and poor water quality posed a severe threat to aquatic ecosystems and created a water resource crisis in the arid areas. The same result of the water quality index and eutrophication also showed the effectiveness of the *CCME WQI* evaluation method.

*4.4. Relationship between Hemeroby Index and Rivers' Environment*

For economic and socio-cultural reasons, human interventions on terrestrial land (such as the spread of chemical substances on land, surface sealing, vegetation cover decrease, and soil erosion) directly impact environmental degradation, water pollution, and biodiversity loss [54]. In recent years with the rapid development of urbanization and industrialization, wastewater discharge, agricultural fertilization, environmental disasters, harvesting of forests, deposition of atmospheric pollutants, and soil erosion increased the content of nutrients in the waterbody [55]. Land-use changes disturbed by human beings affect approximately three-quarters of all vegetated lands, disturbing the nitrogen cycle and availability for use [23]. This environmental degradation has interfered with the natural process of the waterbody and accentuated eutrophication in rivers of Shanxi Province. Considering the eutrophication and completely disturbed were more related, it was very effective to control construction and bare land reasonably in land-use planning [56]. In order to reduce the eutrophication risk of water resources in arid areas, policymakers should: (1) control point source discharges of nitrogen and phosphorus in sewage and domestic treatment plants; (2) control non-point source discharges of nitrogen and phosphorus through mitigating nutrient transfers to surface water including the use of livestock feed and farmland fertilizer; (3) develop sustainable land-use policy characterized by low nutrient exports.

Human beings affect surface water quality by interfering with land-use cover types [57]. Song et al. [58] showed that construction and agricultural lands negatively influenced surface water quality, while forests could benefit water quality by mitigating soil erosion, increasing physical absorption, and intercepting water runoff. The size, density, and diversity of landscapes could influence nutrient transfer, hydrological characteristics, and energy dynamics, impacting water quality [59]. Therefore, the proportion of land-use covers has become an effective means to explain water quality change [24]. For example, land-use and hydrological variables could explain more than 50% of the variation in water quality, of which urban and industrial variables accounted for more than 70% [30].

The southwest of Shanxi Province had high GDP, high population density, concentrated industry development, and seriously affected ecological environment. In addition, due to its proximity to the Yellow River, Shanxi Province's soil erosion was also very large. Altogether, these were the important reasons for the poor water quality. In order to reduce the risk of water quality degradation in arid areas, policymakers should: (1) identify pollutant sources and control pollution through planning human-altered land-use spatial pattern; (2) restore and reinforce the buffer strip areas around strong interference land-use covers (such as agroecosystems, urban and industrial clusters); (3) focus on the proportion of completely interference land-use types by increasing the area of green land and natural water.

**5. Conclusions**

A land-use cover-based anthropogenic disturbance model was used to understand the correlation between anthropogenic activity intensity and rivers' environment in Eastern Loess Plateau. The rivers' status was strongly associated with land-use type. Eutrophication has a greater correlation with fully disturbed land-use types (construction and agricultural land), and the risk of regional water quality degradation is higher when a higher proportion of such land-use types are present. The method used in this study was effective and feasible for assessing the rivers' state under different anthropogenic disturbances. For example, when performing land-use planning, the government can identify the relationship between

the level of regional water quality pollution and human disturbance in relation to historical land-use types. The conclusions reached can provide reference suggestions for the next land planning.

Our results illustrated the correlation between anthropogenic activity and water quality of the rivers on the Loess Plateau and contributed to a better understanding of the ecological effects that anthropogenic activities have on different landscape configurations. This research would provide scientific guidance for water ecosystem management and regional sustainable development for maintaining ecological balance and regulating anthropogenic activities in arid areas.

**Supplementary Materials:** The following are available online at https://www.mdpi.com/article/10.3390/w13223305/s1. Table S1: Classification of land-use cover and hierarchy with respect to Hemeroby index. Table S2: Abbreviation, unit, grading standard, and determination method of 15 hydrochemical indices. Table S3: Anthropogenic interference levels including hemoroby index, completely disturbed, and completely disturbed proportion in different watersheds and rivers.

**Author Contributions:** Conceptualization: L.J. and Y.B. Data curation: L.J. Funding acquisition: L.J. and Y.B. Investigation: Y.L. and G.Z. Supervision: L.J. Writing—original draft: L.J. Writing—review and editing: Y.B. All authors have read and agreed to the published version of the manuscript.

**Funding:** This work was supported by the National Key Research and Development Project (No.2020YFA0907402), National Science and Technology Major Project for Water Pollution Control and Treatment (No.2017zx07108-001), and the grants from CAS Key Laboratory of Soil Environment and Pollution Remediation, Institute of Soil Science, Chinese Academy of Sciences (SEPR2020-03).

**Data Availability Statement:** The data presented in this study are available upon request from the corresponding author.

**Acknowledgments:** We are thankful to Yuan Li, Guixiang Zhang, and Yonghong Bi for their assistance with fieldwork and analysis of water samples. Thanks to all the staff at State Key Laboratory of Freshwater Ecology and Biotechnology, Institute of Hydrobiology (Chinese Academy of Sciences), for their hard work and dedication.

**Conflicts of Interest:** The authors declare that they have no known competing financial interests or personal relationships that could have appeared to influence the work reported in this paper.

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
