# Peer review of "Anthropogenic Disturbances Have Contributed to Degradation of River Water Quality in Arid Areas"

_water, doi:10.3390/w13223305_

Round 1

Reviewer 1 Report

The proposed work is based on a simple but very topical intuition: the increase in population and the increase in anthropogenic activities cause a decrease in the general environmental quality of river ecosystems. However, each proposed tool will be of great importance for identifying procedures for environmental remediation and for raising awareness among policy makers with innovative and functional solutions.
The introduction is well articulated and explanatory, it very well highlights the global context in which it is possible to use the proposed methodology, confirming the importance of this project.

The paragraph on materials and methods is clear, the simplicity of the proposed index has certainly helped to make the method easily understandable. However, increasing the starting database would be of great relevance. 

To test the proposed test on a larger scale, a larger dataset would be needed, on different river environments to strengthen the proposed statistical analysis.

Results and discussions: These two paragraphs need to be divided. The results paragraph has a different function than the discussions.
However, the graphs are well done and well resume the presentation of the results supported by a good statistical analysis, even the discussions are smooth and well articulated.

The article as a whole is well structured starting from a valid project idea of ​​great socio-economic importance. The proposed index has all the capabilities described by the authors to initiate a process of support for institutions to increase sensitivity to environmental matters, in particular to the protection of rivers.

Author Response

Reviewer #1:

  1. Results and discussions: These two paragraphs need to be divided. The results paragraph has a different function than the discussions.

Response: Thanks for your comment. We have separated the results from the discussion and enhanced the discussion section.

Reviewer 2 Report

Major comments:

In this study, the authors have studied “Evaluating spatial changes of human disturbances and their effects on rivers’ environmental quality on The Loess Plateau, China”. The authors discussed the additional spatial information and evidence for the relationship between anthropogenic activities and the river environment. One of the major flaws is that the experimental design and results are rather simple, making the manuscript lack of strong elucidation to support their novelty. The literature review is not strong enough to provide research gaps. The results and discussion section is the most important part of a study, but no critical discussion has been made. As this section progress, there is lacking of connection of results with each other. The English language used in the study needs improvement, as there are punctuation and grammatical mistakes throughout the manuscript. Sentences need more clarity and better construction. Study recent literature and emphasis the latest topics and trends in this region. It is obvious the quality of the manuscript does not meet the standards of the Water Journal, therefore, needs major revisions or should be rejected in its present form. The authors are advised to address the following comments very carefully.

 Introduction:

The introduction is lack of sufficient background information which is unable to give the reader detailed background knowledge and possible wide application of this study. The literature section is poorly written. It needs to be more emphasized on the research work with a detailed explanation of the whole process considering past, present and future scope. It is strongly recommended to add a recent literature survey about environmental pollution, how the current water pollution levels affect the atmosphere and environment. Research gaps should be highlighted more clearly and future applications of this study should be added. The novelty of the study is missing. Research gaps and future applications of the study should be added.

Materials and methods:

This section is also not satisfactory. There are many sentence related mistakes. Many of the sentences are vague and need proper explanation. Terms and units need correction.

 Specific comments:

  1. The start of the abstract is looking weak owing to the unavailability of background knowledge of this study. Add a comprehensive line at the start of the abstract about the background history of the work. Also, add some key values from the results and highlight the novelty of this work clearly. The ending of this section is quite abrupt. Complete the abstract with a conclusive on this work and its findings.
  2. Page 1, line 21: “a significant negative correlation with HI”. What does CDI stand for?
  3. Introduction and background is weak, no strong information is provided about the different types of environmental pollutions their effects on humans such as from wastewater industries the major contributor in environmental pollution, therefore, the authors are advised to read and add environmental pollution types and levels from the following studies: Case Studies in Chemical and Environmental Engineering, 2020; 2:100037. Journal of Environmental Management, 2020; 278:111302. Environmental Research, 2021;195:110839. Journal of Environmental Chemical Engineering, 2021;9(1):104502. Chemosphere, 2021;270:129523.
  4. Page1, line 40-41: “anthropogenic activities have led to irreversible changes in landscape structure.” Please mention the detail of the anthropogenic activities.
  5. Page 1, line 21: “which have generated enormous impacts on river ecosystems”. Explain the impacts on river ecosystems with mechanism clearly.
  6. Page2, line 56: “Three Gorges basin”. Please discuss the three gorges basin properly.
  7. Page 2, line 90: “ecological vulnerability”. Briefly define this term with literature.
  8. Page 4, line 78-79: “Quantitative evaluation of regional anthropogenic activities would help us get insight into the relationship between human activities and the water environment”. How would the quantitative evaluation of regional anthropogenic activities help us get insight into the relationship between human activities and the water environment? Critically discuss with sated values from literature.
  9. Page3, line 105-106: “We analyzed the characteristics of different land-use covers and quantified the intensity of anthropogenic disturbance by adopting remote sensing and field investigation”. This sentence is vague. Rewrite it comprehensively.
  10. Page 5, line 175: “Environment Water Quality Index”. Describe this term clearly with mechanism.
  11. Page3, line 125-126: “Shanxi Province has more than 1000 large and small rivers with noticeable seasonal variation in water quantity and 250 rivers with a drainage area of more than 100 km2.” Please add the latest reference in this sentence.
  12. More recent research about types of wastewater pollution, treatment methods and pollution reduction technologies is suggested to be added to make the background and discussion more strong: Asia‐Pacific Journal of Chemical Engineering, 2016; 11(6):855-65. Journal of Environmental Chemical Engineering, 2021;9(1):104502. Chemosphere, 2022;287: 132114. Journal of the Taiwan Institute of Chemical Engineers, 2021;125:141-152. Journal of Molecular Liquids, 2021;335:116567.

  1. Page 5, line 175-176: “the exceeding range, the exceeding frequency, and the exceeding range of pollutants". Please explain these three aspects step by step clearly.
  2. Page 5, line 179: What does “chla” mean in this line?
  3. Page 7, line 230: “The maximum of TN” Explain this term completely.
  4. Page 6, line 401: “GDP” Write the full form of the abbreviation at first sight.
  5. Page 6, line 414-415: "The eutrophication had a much stronger relationship with the completely disturbed HI, and water quality was most likely degraded when the completely disturbed land-use type occupied a higher proportion". Rewrite it properly.
  6. The conclusions only talk about some studied parameters, which is insufficient to depict the whole picture of the contribution of this study. The authors are advised to write the conclusions in a comprehensive way and should contain key values, suitability of the applied method, the major findings, contributions and possible future work.
  7. The authors are advised to revise references, including the latest references. Please see some suggestions in the specific comments.

Author Response

Reviewer #2:

  1. The start of the abstract is looking weak owing to the unavailability of background knowledge of this study. Page 1, line 21: “a significant negative correlation with HI”. What does CDI stand for?

Response: Thanks for your comment. We have rewritten the introduction section.

“The earth has been reshaped for millennia. The accelerating pace of anthropogenic activities has generated enormous impacts on water environment. As one of the main drivers of landscape change, anthropogenic disturbance has brought many negative effects on rivers. Studying the relationship between anthropogenic disturbances and river water quality is of significance for regional conservation and ecosystem management, whilst, the relationship remains poorly understood in the current. In this study, we quantified anthropogenic disturbances by introducing the concept of hemeroby index and evaluated rivers’ water quality in eight sub-watersheds on the Loess Plateau. The results indicated that 37.5% of the sub-watersheds was in Eutrophic status and 62.5% was in Marginal water quality index. The river water quality was most poor in the south-western region near the Yellow River with high-level anthropogenic disturbance. A correlation analysis between water quality indicators and hemeroby suggested that anthropogenic disturbance contributed to a significant water quality deterioration trend (p < 0.01). The river water quality was relatively sensitive to the changes of completely disturbed land-use covers including urban and industrial land. Our findings provide theoretical guidance for regional water resources conservation and ecosystem management in arid area.”

  1. Page1, line 40-41: “anthropogenic activities have led to irreversible changes in landscape structure.” Please mention the detail of the anthropogenic activities. Page 1, line 21: “which have generated enormous impacts on river ecosystems”. Explain the impacts on river ecosystems with mechanism clearly.

Response: Thanks for your comment. We have added explanations in line 49-58, Page 4.

“On the other hand, anthropogenic activities (excessive pollution discharge, nutrient pollution from increased construction land, agricultural surface source pollution, and the industry discharge including textile industry, metal mining industry, and pharmaceutical industry etc.) (Rasheed et al., 2020; Rashid et al., 2020; Sadiq et al., 2021;) have led to irreversible changes in landscape structure, which have generated enormous impacts on river ecosystems (Ning et al., 2015). The deterioration of water quality has constrained the socio-economic development, even posing a severe threat to the ecosystem, food safety, and human health (Bagheri et al., 2021; Duan et al. 2016; Zhang et al., 2019), which has become a common problem facing all countries in the world (Yang & Chen, 2020).”

  1. the authors are advised to read and add environmental pollution types and levels from the following studies.

Response: Thanks for your comment. We have added some relevant literature to illustrate the impact of wastewater pollutants in introduction.

Bagheri, A., Aramesh, N., Sher, F., & Bilal, M. (2021). Covalent organic frameworks as robust materials for sustainable mitigation of environmental pollutants. Chemosphere, 270, 129523. https://doi.org/10.1016/j.chemosphere.2020.129523

Rasheed, T., Ahmad, N., Nawaz, S., & Sher, F. (2020). Photocatalytic and adsorptive remediation of hazardous environmental pollutants by hybrid nanocomposites. Case Studies in Chemical and Environmental Engineering, 2, 100037. https://doi.org/10.1016/j.cscee.2020.100037

Rashid, T. Sher, F., Hazafa, A., Hashmi, R., Zafar, A., Rasheed, T., & Hussain, S. (2020). Design and feasibility study of novel paraboloid graphite based microbial fuel cell for bioelectrogenesis and pharmaceutical wastewater treatment. Journal of Environmental Chemical Engineering, 9, 104502. https://doi.org/10.1016/j.jece.2020.104502

Sadiq, H., Sher, F., Sehar, S., Lima, E., Zhang, S., Iqbal, H. Zafar, F., & Nuhanovic, M. (2021). Green synthesis of ZnO nanoparticles from Syzygium cumini leaves extract with robust photocatalysis applications. Journal of Molecular Liquids, 335, 116567. https://doi.org/10.1016/j.molliq.2021.116567

Sher, F., Hanif, K., Rafey, A., Khalid, U., Zafar, A., Mariam, A., Lima, E. (2020). Removal of micropollutants from municipal wastewater using different types of activated carbons. Journal of Environmental Management, 278, 111302. https://doi.org/10.1016/j.jenvman.2020.111302

  1. Page2, line 56: “Three Gorges basin”. Please discuss the three gorges basin properly.

Response: Thanks for your comment. This is our mistake for expression. We have modified it in line 68-70, Page 4-5.

  1. Page 2, line 90: “ecological vulnerability”. Briefly define this term with literature.

Response: Thanks for your comment. We have modified it in line 106-109, Page 6.

“Yang & Song (2021) assessed the spatio-temporal characteristics of ecological vulnerability (sensitivity and recover ability of ecosystems to external disturbances) based on the HI and found that the ecological environment in high-disturbance areas was more vulnerable than low disturbance areas.”

  1. Page 4, line 78-79: “Quantitative evaluation of regional anthropogenic activities would help us get insight into the relationship between human activities and the water environment”. How would the quantitative evaluation of regional anthropogenic activities help us get insight into the relationship between human activities and the water environment? Critically discuss with sated values from literature.

Response: Thanks for your comment. We have modified it in line 94-98, Page 6.

“Quantitative evaluation of regional anthropogenic activities would help us get insight into impacts from human.  A correlation analysis between landscape indices and hemeroby suggested that the landscape patterns in regions with high-level human disturbance were relatively sensitive to water quality and species richness (Zhou et al., 2018).”

  1. Page3, line 105-106: “We analyzed the characteristics of different land-use covers and quantified the intensity of anthropogenic disturbance by adopting remote sensing and field investigation”. This sentence is vague. Rewrite it comprehensively.

Response: Thanks for your comment. We have modified it in line 123-126, Page 7.

“We analyzed the characteristics of different land-use covers and quantified the intensity of anthropogenic disturbance using remote sensing data and geographical information system technology.”

  1. Page3, line 125-126: “Shanxi Province has more than 1000 large and small rivers with noticeable seasonal variation in water quantity and 250 rivers with a drainage area of more than 100 km2.” Please add the latest reference in this sentence.

Response: Thanks for your comment. These figures were taken from the government website (http://www.shanxi.gov.cn/) and we have made changes. We have modified it in line 144-146, Page 7-8.

  1. Page 5, line 175: “Environment Water Quality Index”. Describe this term clearly with mechanism. Page 5, line 175-176: “the exceeding range, the exceeding frequency, and the exceeding range of pollutants". Please explain these three aspects step by step clearly.

Response: Thanks for your comment. This is our mistake for expression. We have modified it in line 194-201, Page 10.

“Water quality index (WQI) is a kind of environmental quality index, it is a combination of several water quality parameters of the test results, to describe the quality of water quality of a comprehensive dimensionless value. The Canadian Council of Ministers of the environment water quality index (CCME WQI) is a tool for measuring the pollution status of a waterbody from three aspects: quantity, frequency, and extent of pollutants exceeded water quality standards which reflect the comprehensive situation of water environment succinctly and intuitively (CCME, 2013).”

  1. Page 5, line 179: What does “chla” mean in this line?; Page 7, line 230: “The maximum of TN” Explain this term completely; Page 6, line 401: “GDP” Write the full form of the abbreviation at first sight.

Response: Thanks for your comment. These are our mistakes. We have modified it in line 201, 213, Page 10 and line 365, Page 17.

  1. Page 6, line 414-415: "The eutrophication had a much stronger relationship with the completely disturbed HI, and water quality was most likely degraded when the completely disturbed land-use type occupied a higher proportion". Rewrite it properly. The conclusions only talk about some studied parameters, which is insufficient to depict the whole picture of the contribution of this study. The authors are advised to write the conclusions in a comprehensive way and should contain key values, suitability of the applied method, the major findings, contributions and possible future work.

Response: Thanks for your comment. We have modified it in line 441-451, Page 20.

“A land-use cover-based anthropogenic disturbance model was used to understand the correlation between anthropogenic activity intensity and rivers’ environment in Eastern Loess Plateau. The rivers’ status was strongly associated with land-use type. Eutrophication has a greater correlation with fully disturbed land-use types (construction and agricultural land), and the risk of regional water quality degradation is higher when a higher proportion of such land-use types are present. The method used in this study was effective and feasible for assessing the rivers’ state under different anthropogenic disturbances. For example, when making land-use planning, the government can identify the relationship between the level of regional water quality pollution and human disturbance in relation to historical land-use types. The conclusions reached can provide reference suggestions for next land-planning.”

Reviewer 3 Report

General comment  - work done very well and carefully, and is an interesting research paper that provides information on the river status and relations between anthropogenic activities and water ecosystem management.

While the analyzed samples were fairly small, it provides some good initial research results understanding the ecological load that human activities cause and their impacts on the environment and water quality.

I am delighted to see that such extensive research is still carried out.

The question arises, how does this result translate into practice? Can you please explain briefly in conclusions? 

Author Response

Dear editors and reviewers:

Thank you for giving us the opportunity to submit a revised draft of the manuscript “Evaluating spatial changes of human disturbances and their effects on rivers’ environmental quality on The Loess Plateau, China” (water-1439117) for publication in the Journal of Water. We appreciate the time and effort that you and the reviewers dedicated to providing feedback on our manuscript and are grateful for the insightful comments on and valuable improvements to our paper. We have incorporated most of the suggestions made by the reviewers. Those changes are highlighted in the manuscript. Please see below, in blue, for a point-by-point response to the reviewers’ comments and concerns. All page numbers refer to the revised manuscript file with tracked changes.

Replies to the reviewers’ comments:

Reviewer #3:

  1. The question arises, how does this result translate into practice? Can you please explain briefly in conclusions?

Response: Thanks for your comment. We have modified it in line 441-451, Page 20.

“A land-use cover-based anthropogenic disturbance model was used to understand the correlation between anthropogenic activity intensity and rivers’ environment in Eastern Loess Plateau. The rivers’ status was strongly associated with land-use type. Eutrophication has a greater correlation with fully disturbed land-use types (construction and agricultural land), and the risk of regional water quality degradation is higher when a higher proportion of such land-use types are present. The method used in this study was effective and feasible for assessing the rivers’ state under different anthropogenic disturbances. For example, when making land-use planning, the government can identify the relationship between the level of regional water quality pollution and human disturbance in relation to historical land-use types. The conclusions reached can provide reference suggestions for next land-planning.”

Reviewer 4 Report

Ji and co-authors present a quantitative model exploring the correlation between human disturbance and rivers trophic state by integrating remote sensing data and hydro-chemical indices. The outcomes are interesting and can contribute to advancing our knowledge in preventing and monitoring eutrophication in areas suffering from multiple anthropogenic pressures. The manuscript is well structured, and the result are soundly presented, however my main criticism concerns the following points:

  • What does the model adopted here add to other prior works? How does this work advance nutrient management decision-making beyond other existing models previously tested and applied? The main novelty of the work and its additional value to the literature is not very clear starting from the abstract and throughout the paper.
  • as it stands, the paper reads like a regional technical report about trophic state evaluation. I think there needs to be a partial rewrite of the introduction, discussion, and conclusions to convert the manuscript into a more interesting scientific paper.
  • I suggest to partially revise the Introduction and also the Discussion making them more interesting for an international audience and not focused only on China. At present, almost the entire Introduction deals with Chinese problems linked to nitrogen contamination, otherwise these common problems worldwide. Same considerations are valid for the Discussion.
  • The paper has a decidedly geologic-hydrologic approach and some additional discussion of the role of biological processes responsible for nutrient retention or transformation would greatly expand the value of the study.
  • A deeper and speculative interpretation of the presented data is needed, in a way to better evaluate both the scientific level of the presented experimental approach, and also their spendibility for practical application in management programmes of agricultural watersheds worldwide and not limited to China.

In addition, a review of grammar and spelling throughout the paper is needed.

Author Response

Reviewer #4:

  1. I suggest to partially revise the Introduction and also the Discussion making them more interesting for an international audience and not focused only on China. At present, almost the entire Introduction deals with Chinese problems linked to nitrogen contamination, otherwise these common problems worldwide. Same considerations are valid for the Discussion. The paper has a decidedly geologic-hydrologic approach and some additional discussion of the role of biological processes responsible for nutrient retention or transformation would greatly expand the value of the study.

A deeper and speculative interpretation of the presented data is needed, in a way to better evaluate both the scientific level of the presented experimental approach, and also their spendibility for practical application in management programmes of agricultural watersheds worldwide and not limited to China. In addition, a review of grammar and spelling throughout the paper is needed.

Response: Thanks for your comment. We have rewritten introduction and discussion in this manuscript. Some important changes are as follows:

Introduction

in line 49-58, Page 4.

“On the other hand, anthropogenic activities (excessive pollution discharge, nutrient pollution from increased construction land, agricultural surface source pollution, and the industry discharge including textile industry, metal mining industry, and pharmaceutical industry etc.) (Rasheed et al., 2020; Rashid et al., 2020; Sadiq et al., 2021;) have led to irreversible changes in landscape structure, which have generated enormous impacts on river ecosystems (Ning et al., 2015). The deterioration of water quality has constrained the socio-economic development, even posing a severe threat to the ecosystem, food safety, and human health (Bagheri et al., 2021; Duan et al. 2016; Zhang et al., 2019), which has become a common problem facing all countries in the world (Yang & Chen, 2020).”

in line 68-76, Page 4-5.

One study found that anthropogenic fertilizer inputs and wastewater discharge were the leading cause of eutrophication in the upstream of the Three Gorges Reservoir (Strokal et al., 2020), and the surface water quality was closely related to land use type (Gu et al., 2019). Another study analyzed the relationship between anthropogenic activities and hydrochemical indices of the Fen River Basin; its results demonstrated that primary pollution sources were related to the land-use pattern of the high proportions of the cropland and the low proportions of the forest (Chai et al., 2020). Even many microscopic pollutants that are difficult to identify are already discharged into rivers and reservoirs (Sher et al., 2020).

in line 94-98, Page 5-6.

“Quantitative evaluation of regional anthropogenic activities would help us get insight into impacts from human. A correlation analysis between landscape indices and hemeroby suggested that the landscape patterns in regions with high-level human disturbance were relatively sensitive to water quality and species richness (Zhou et al., 2018).”

in line104-109, Page 6.

“Yang & Song (2021) assessed the spatio-temporal characteristics of ecological vulnerability (sensitivity and recover ability of ecosystems to external disturbances) based on the HI and found that the ecological environment in high-disturbance areas was more vulnerable than low disturbance areas.”

in line123-128, Page 7.

“We analyzed the characteristics of different land-use covers and quantified the intensity of anthropogenic disturbance using remote sensing data and geographical information system technology. Then we evaluated the eutrophication status and comprehensive index of surface water in different watersheds under different anthropogenic disturbance backgrounds.”

Discussion

We have redone the discussion in the following four main areas:

4.1 Spatial variation of hydrochemical indices

4.2 Anthropogenic disturbances of different sub-watersheds

4.3 Rivers’ state of different watersheds and sub-watersheds

4.4 Relationship between Hemeroby index and rivers’ environment

Round 2

Reviewer 2 Report

The authors have responsed well to those proposed questions by reviewers and carefully revised the whole paper according to the suggestions of reviewers step by step. Therefore, the quality of this paper has elevated greatly, hence I agree to publish the current version of the paper without any further changes.

Author Response

Thank you for your review, it is a great help to enhance and improve the manuscript

Reviewer 4 Report

The authors have greatly improved the manuscript.

Some last minor comments:

  • line 75: not clear what are "microscopic pollutants". Please consider revise.
  • The authors should stress more the novelty of their work.

Author Response

Thanks for your comment. We have added the specific descriptions about  microscopic pollutants on Page 5, Line 75-77. "Even many microscopic pollutants, such as pharmaceuticals, organic polymers, and suspended solids, that are difficult to identify are already discharged into rivers and reservoirs ".

We summarize the latest research developments and results, and point out specific problems, followed by highlighting the main work of this paper to solve this problem on Page 7, Line 123-126."However, current research rarely addresses the quantified impacts of human activities on the aquatic environment at a large scale. To address this issue, we developed a strategy to quantify the impact of human activities on surface water through a case study of a typical arid area, the Loess Plateau of northern China."
